# Design and Study of Physical and Mechanical Properties of Concrete Based on Ferrochrome Slag and Its Mechanism Analysis

**Meiyan Hang, Jiechao Wang * , Xuebin Zhou and Mengjie Sun**

School of Civil Engineering, Inner Mongolia University of Science and Technology, Baotou 014000, China
* Correspondence: kneb13042650185@163.com; Tel.: +86-13042650185

**Abstract:** In this study, high-carbon ferrochrome slag powder produced by grinding was used to replace different proportions of cement, and the effect of the amount of ferrochrome slag powder on the physical and mechanical properties of ferrochrome-slag-cement composites was analyzed. Water-cooled ferrochrome slag with a particle size of <5 mm was optimized to replace part of river sand as fine aggregate, and air-cooled ferrochrome slag with a particle size of >5 mm was used to completely replace coarse aggregate to prepare ferrochrome-slag-based concretes. The microstructure of ferrochrome-slag-cement composites was analyzed by X-ray diffraction, scanning electron microscopy, and thermogravimetry–differential scanning calorimetry analysis. The compressive strength, water absorption, and aggregate–slurry interface bonding properties of ferrochrome-slag-based concrete were studied. The results demonstrate that a ferrochrome slag powder amount of 15% leads to the highest performance of ferrochrome-slag-cement composite material, and the fluidity ratio of ferrochrome-slag-cement mortar is 103, higher than reference samples. Furthermore, the compressive strengths of ferrochrome slag concretes are 15.8% and 3.6% higher than conventional concrete, and the water absorption of ferrochrome slag low-carbon concrete is better than that of conventional concrete. The interface bonding structure between concrete aggregate and slurry was optimized. This research can provide a reference for studying the application of ferrochrome slag, both the feasibility of high-carbon ferrochrome slag powder as supplementary cementitious material and the application of ferrochrome slag as concrete aggregate, and it can help to achieve the purpose of saving energy and reducing carbon emissions.

**Keywords:** ferrochrome slag; compressive strength; microstructure; concrete aggregate; interface bonding

## 1. Introduction

In recent years, the rapid growth of the human population, and the rapid development of industrialization and urbanization, have increased the amount of concrete consumed by the construction industry and have put great pressure on the development of the construction industry. In recent years, trying to find a path for the sustainable development of the construction industry has become a focus of attention [1–3]. Concrete is the most commonly used building material in the world because of its low price, availability of raw materials, strength, durability, and fire resistance [3].

The main materials of traditional concrete are natural aggregate and cement. Aggregates account for approximately 70–85% of the total of concrete. Due to the development of the global construction industry, more than 50 billion tons of aggregate are mined every year. However, the huge demand for natural aggregate resources may lead to great pressure to obtain raw materials in the production of concrete. In addition, the use of cement in concrete is a key factor in the increase of greenhouse gas emissions. Each ton of cement clinker produced emits approximately 1.2 tons of carbon dioxide. This has created huge

pressure for the sustainable development of the construction industry. According to the International Energy Agency and the Cement Sustainability Initiative, cement production may increase by 23% by 2050. Therefore, promoting eco-sustainable materials in the concrete industry is critical to the long-term sustainability of the construction industry and the efficient use of natural resources [4].

The use of industrial solid waste and construction and demolition waste has always been a challenge in industrial and construction development [2,3]. In recent decades, efforts have been made to incorporate industrial solid waste and construction and demolition waste into the development of building materials, to reduce the pressure on the development of the construction industry, and to reconcile the development of the concrete industry with environment protection. The used of industrial waste as concrete aggregate or supplementary cementitious material is an effective way to obtain sustainable resources. Xie H Z et al. [2] used waste concrete powder (WCP) by grinding before use. The interconnected porosity, permeability, and strength of concrete mixtures with 100% recycled coarse aggregate and different amounts of WCP were tested. The results showed that adding WCP instead of paste has great application potential in the production of environmentally friendly high-performance pervious concrete.

Ferrochrome slag is the residue produced in the production process of ferrochrome alloy [4,5]. Each ton of ferrochrome alloy produces approximately 1.1–1.6 tons of ferrochrome slag. The chemical composition of ferrochrome slag is mainly $SiO_2$, $Al_2O_3$, and $MgO$. Its mineral composition is mainly magnesium aluminate spinel and magnesium chromite, chromate, glassy phase, and a small amount of other minerals. Most ferrochrome slag is landfilled or stacked, and the resource utilization rate is only 30% [6,7]. This not only pollutes groundwater and the natural environment, but also occupies a lot of land.

At present, ferrochrome slag is mainly used as concrete aggregate in the field of building materials. Fares A I, et al. [8] reviewed the physical and mechanical properties of ferrochrome slag, and they discussed its application as fine and coarse aggregates in sustainable green concrete production. The results indicated that the physical and mechanical properties of ferrochrome slag aggregate are generally superior to those of conventional aggregate. Das S et al. [9] used ferrochrome slag to partially replace concrete fine aggregate and rice husk ash to replace cement to produce sustainable concrete. The results demonstrated that the use of ferrochrome slag instead of natural sand had no significant effect on the performance of concrete. When a higher proportion of rice husk ash was used to replace cement, the performance of the concrete decreased.

In another study, Prasanna K A et al. [10] discussed the feasibility of using ferrochrome ash as a supplementary cementitious material to produce environmentally friendly concrete. Ferrochrome ash was modified with lime powder to improve the performance of concrete. The results demonstrated that the use of ferrochrome ash in the preparation of concrete helped reduce the amount of Ordinary Portland cement, minimize greenhouse gas emissions, and reduce energy consumption. Mehmet B K et al. [11] discussed the effects of alkali dosage and the silicic modulus of sodium metasilicate solution on the polymerization of ferrochrome slag (FS) under different curing conditions. The setting time, hydration heat, and compressive strength of geopolymer paste samples and the compressive strength of geopolymer mortar samples were obtained. When the w/b (water/binder) ratio increased, the compressive strength of the material decreased. The highest 28-day strength of geopolymer mortar was at 0.30 water–binder ratio and laboratory temperature.

In this context, the present study focused on ferrochrome slag as a supplementary cementitious material and concrete aggregate. The utilization of industrial by-products and solid waste can contribute to the sustainable development of building materials. Previous research has used ferrochrome ash and low-carbon ferrochrome slag powder as supplementary cementitious material; however, there are few studies on using high-carbon ferrochrome slag as supplementary cementitious material. Therefore, this paper introduces the feasibility of water-cooled high-carbon ferrochrome slag after grinding as a supplementary cementitious material. The effects of the amount of water-cooled

ferrochrome slag powder on the physical and mechanical properties of cement and concrete were studied. The determination of an optimal substitution rate for water-cooled high-carbon ferrochrome slag powder as a supplementary cementitious material was made, and this provides a reference for the use of water-cooled high-carbon ferrochrome slag as supplementary cementitious material in practical engineering.

Previous research has mainly focused on the influence of using water-cooled ferrochrome slag as fine aggregate and natural coarse aggregate on the physical and mechanical properties of concrete, or the effects of using air-cooled ferrochrome slag as coarse aggregate and natural fine aggregate on the physical and mechanical properties of concrete. The effect of using ferrochrome slag as fine aggregate and air-cooled ferrochrome slag as coarse aggregate on the basic properties of concrete is rarely reported. Therefore, this paper introduces ferrochrome slag as concrete aggregate, the preparation of ferrochrome-slag-based concrete by using ferrochrome slag powder as cementitious material to replace part of the cement, water-cooled ferrochrome slag to replace 40% of river sand as fine aggregate of concrete, and air-cooled chromite slag as coarse aggregate [12]. The effects on compressive strength, water absorption, and aggregate–slurry interface bonding properties of ferrochrome-slag-based concrete are studied. The aim is to provide a reference for the application of both as concrete aggregates in practical engineering, and achieve the effective use of industrial solid waste, protect the natural environment, and reduce carbon emissions.

## 2. Materials and Methods

### 2.1. Materials

#### 2.1.1. Cement

The P·O 42.5 grade Ordinary Portland cement was produced by Inner Mongolia Meng xi Co., Ltd., Ordos, China. The main chemical composition is shown in Table 1, and the performance and property parameters of the cement are presented in Table 2.

**Table 1.** Chemical constituents of cement and ferrochrome slag.

| Constituents | $SiO_2$ | $Al_2O_3$ | $Fe_2O_3$ | CaO | MgO | $SO_3$ | $Cr_2O_3$ |
|---|---|---|---|---|---|---|---|
| Cement | 26.30% | 9.63% | 3.69% | 55.43% | 2.10% | 1.23% | - |
| Ferrochrome slag | 28.26% | 32.31% | 3.08% | 3.34% | 27.62% | 0.75% | 3.82% |

**Table 2.** Physical properties of OPC.

| | Initial Setting Time/(min) | Final Setting Time/(min) | Specific Surface/(m²/kg) | Consistency /% | Compressive Strength/MPa | |
|---|---|---|---|---|---|---|
| | | | | | 3 d | 28 d |
| OPC | 164 | 220 | 382.4 | 25.4 | 32.7 | 48.3 |

#### 2.1.2. Ferrochrome Slag Powder

The ferrochrome slag powder used in this investigation was collected from the Ming Tuo Group Co., Ltd., located in Baotou, Inner Mongolia Province, China. Water-cooled ferrochrome slag was subjected to ball mill grinding to reduce its particle size to less than 80 μm to obtain ferrochrome slag powder. According to the ferrochrome slag powder particle size distribution analysis, the median particle size ($D_{50}$) was 6.09 μm, and the specific surface area was 640.2 $m^2 \cdot kg^{-1}$. The microstructure of the ferrochrome slag powder is illustrated in Figure 1, while the main chemical composition is presented in Table 1. X-ray diffraction (XRD) analysis of the ferrochrome slag powder is provided in Figure 2. It can be seen that the main mineral composition of the ferrochrome slag powder was magnesium olivine, magnesium aluminum spinel, enstatite, and glass phase [13,14]. The glass phase containing $SiO_2$ and $Al_2O_3$ is the main source of the activity of ferrochrome slag powder.

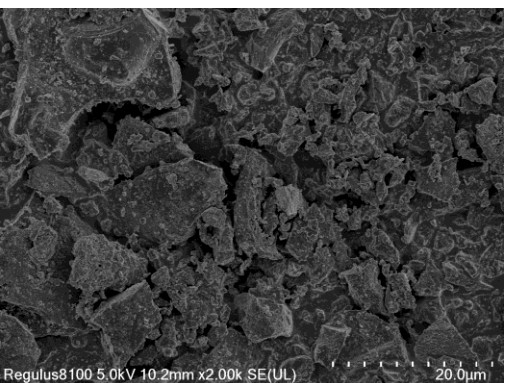

**Figure 1.** Micro-morphology diagram of ferrochrome slag powder.

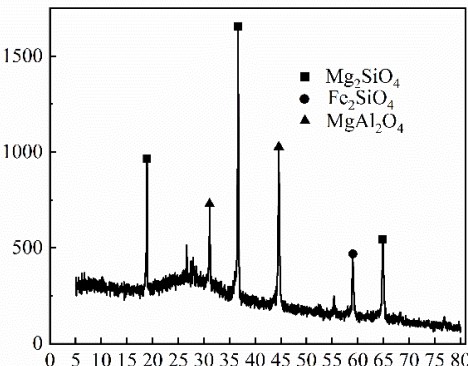

**Figure 2.** XRD of ferrochrome slag.

### 2.1.3. Ferrochrome Slag Fine Aggregate

Using the Baotou Ming tuo Group water-cooled ferrochrome slag, the particle structure and morphology of the single-particle ferrochrome slag were observed by laser focus scanning, and the results are presented in Figures 3 and 4. These figures reveal that the color of water-cooled ferrochrome slag was dark green, and that the structure of the surface was porous and irregular. The physical properties of water-cooled ferrochrome slag are provided in Table 3. The fineness modulus of fine ferrochrome slag was 3.6, and the porosity was 49%, and its grading belonged to zone-I. Therefore, according to the particle size distribution curve of GB/T 14684-2011 grading zone-II, the fine ferrochrome slag was optimized to belong to the zone-II. The optimized fine ferrochrome slag cumulative screening curve is presented in Figure 5.

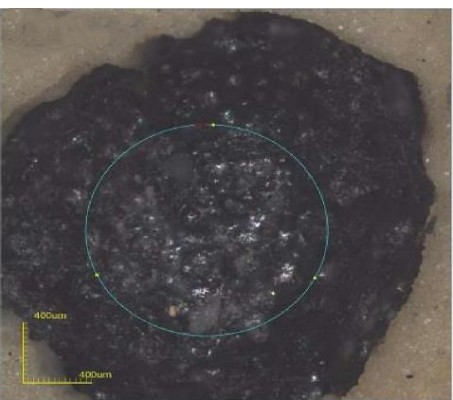

**Figure 3.** Scanning topography of ferrochrome slag by laser confocal electron microscope.

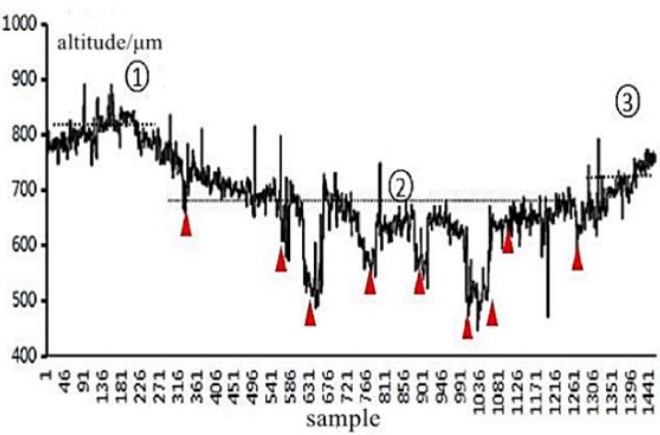

**Figure 4.** The height curve of the appearance sample point of ferrochrome slag.

**Table 3.** Physical properties of fine aggregates.

| | Specific Gravity/Kg·m⁻³ | Bulk Density/Kg·m⁻³ | Porosity /% | Fineness Modulus | Water Absorption /% |
|---|---|---|---|---|---|
| Natural sand | 2609 | 1620 | 37.9 | 2.6 | 0.8 |
| FeCr slag | 2836 | 1447 | 49 | 3.6 | 3.1 |

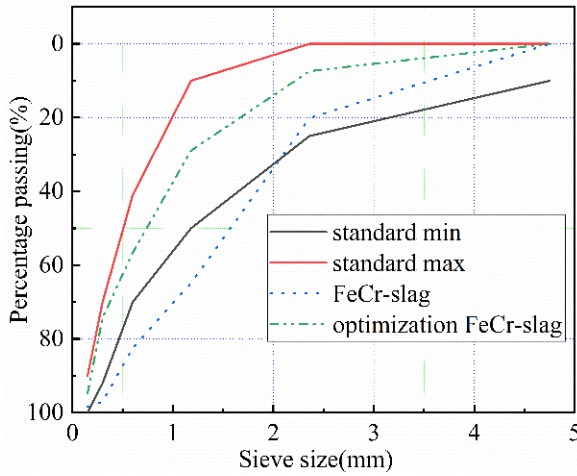

**Figure 5.** Particle size distribution of FeCr slag and Optimization FeCr-slag.

### 2.1.4. Natural Fine Aggregates

River sand collected from a local river, with the size varying between 0.15 mm to 2.36 mm, was used as a fine aggregate; its physical properties are shown in Table 3.

### 2.1.5. Coarse Aggregate

Crushed basalt rock (maximum size of 20 mm) meeting the standard GB/T 14685-2011 was used as coarse aggregate. Crushed air-cooled ferrochrome slag with maximum size of 20 mm (obtained from Ming tuo Group Alloys Ltd., Baotou, China) was utilized as coarse aggregate in concrete manufacturing. The appearance of the coarse aggregate is presented in Figure 6. This figure shows that the coarse ferrochrome slag had a gray-black porous structure, and the physical properties of the slag are provided in Table 4.

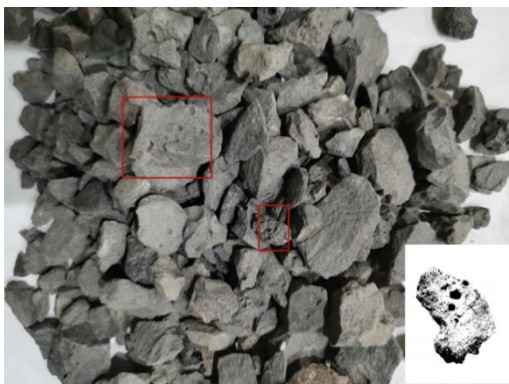

**Figure 6.** Ferrochrome slag after natural cooling by air.

**Table 4.** Physical performance index of coarse aggregate.

| | Specific Gravity/Kg·m$^{-3}$ | Bulk Density/Kg m$^{-3}$ | Porosity % | Gradation | Crushing Value/% |
|---|---|---|---|---|---|
| NCA | 2780 | 1660 | 40.3 | continuous | 8 |
| FeCr slag | 3073 | 1487 | 51.6 | continuous | 12.1 |

2.1.6. Water and Superplasticizer

To mix concrete and cure the specimens, drinkable tap water available in the concrete laboratory was used. To maintain workability, a commercially available polycarboxylic-based superplasticizer was used in all mixes.

*2.2. Experimental Methods*

The cement mortar specimens were mixed with a water–binder ratio and a cement–sand ratio of 1:2 and 1:2.5, respectively. Cement was substituted at five percentages (0%, 10%, 20%, 30%, and 40%) of ferrochrome slag powder by weight in all mortar mixtures. The size of each specimen was 40 mm × 40 mm × 160 mm. The mortar specimens were cured in a mold for a 24 h period, then removed from the mold and the curing was continued for 28 d. According to the standard GB17671-1999, the fluidity and compressive strength were tested at 3, 7, and 28 d.

2.2.1. Mixture Design and Specimen Preparation

Conventional concrete mixes were designed according to JGJ 55-2011 at 50 MPa cube strength. The concrete mix was 1:1.91:2.75 (cement: fine aggregate: coarse aggregate) by weight with a fixed water–cement ratio of 0.40 for all three mixtures. According to the previous test results, replacing 40% of natural river sand with chromite slag sand leads to the best performance of concrete; therefore, the content of ferrochrome slag sand in the ferrochrome-slag-based concrete was determined to be 40% of the total amount of fine aggregate. The slump of fresh concrete was recorded as per GB/T 50081-2019 to examine the impact of ferrochrome slag on the workability of concrete. Concrete cubes 150 mm × 150 mm × 150 mm were produced for compressive strength testing, and the loading rate was 0.5 MPa/s as per GB/T 50081-2019. Concrete cubes 100 mm × 100 mm × 100 mm were produced for water absorption testing. The specimens were cured in water for 24 h and then tested at room temperature at 7, 28, and 56 d. A water absorption test was conducted as per GB/T 50081-2019 on 100 mm × 100 mm × 100 mm concrete cubes after 28 days of curing. The water absorption (W) was calculated by using Equation (1):

$$W = (Ms - Md)/Md \times 100\% \tag{1}$$

where W = water absorption of concrete, Ms = mass of saturated concrete, and Md = mass of dried concrete.

### 2.2.2. X-ray Diffraction Study

The powder mortar samples were studied in a powder X-ray diffractometer (AD-VANCE) utilizing Cu kα radiation. The X-ray diffraction (XRD) spectrum was taken in the 2θ range. The peak position of the spectrum was analyzed, and the mineral phase in the sample was discovered.

### 2.2.3. Thermogravimetric and Differential Scanning Calorimetric Analyses

TG and DSC analyses were performed using a NETZSCH, STA-449C Jupiter analyzer. The samples were placed in a container, and the experiments were conducted in an $N_2$ atmosphere. The heating rate was 10°C/min.

### 2.2.4. Scanning Electron Microscopy Analysis

The scanning electron microscopy (SEM) analysis used the FEI-INSPECT-F50 model microscope. SEM analysis was performed on 28-days-cured samples and ferrochrome concrete samples were compared to conventional concrete samples.

## 3. Results

### 3.1. Fluidity and Compressive Strength of Cement Mortar

The test results for the fluidity and compressive strength of cement mortars with different substitution amounts of chromite slag powder are provided in Figures 7 and 8.

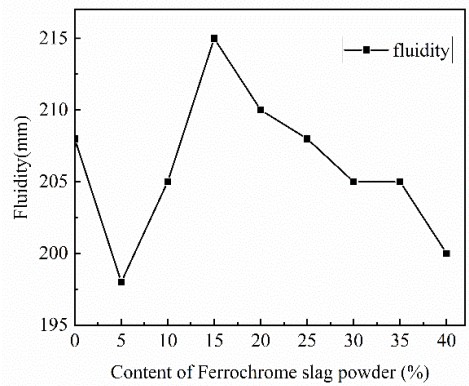

**Figure 7.** Flow chart of cement mortar.

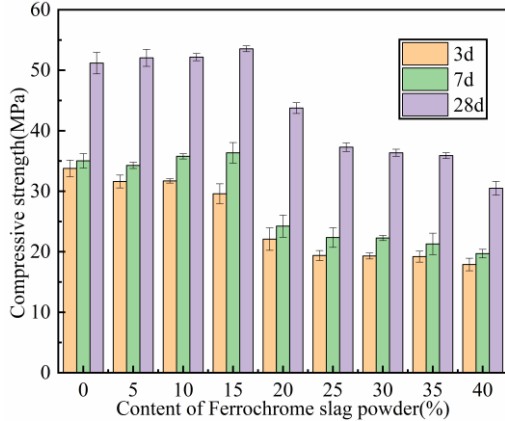

**Figure 8.** Compressive strength of cement mortar.

From Figure 7, it can be seen that with the increase in the substitution amount of ferrochrome slag powder, the fluidity performance of cement mortar can be divided into

three stages. Namely, it decreases in the first stage, increases in the second, and then decreases in the third stage. When the substitution amount of ferrochrome slag powder is 15%, the fluidity of cement mortar is the highest, and the fluidity ratio is 103. With a low substitution amount of ferrochrome slag powder, the fluidity in the first stage decreases, due to the large specific surface area of ferrochrome slag powder, the high content of fine powder, and the water demand of the ferrochrome-slag-cement composite material. With an increase in the substitution amount of ferrochrome slag powder, due to the porous structure of ferrochrome slag powder and its water-reducing effect, ferrochrome slag powder leads to a water absorption-loss process in cement mortar. When initially adding water, the ferrochrome slag powder absorbs water, which increases the viscosity of the cement mortar. After stirring for a period of time, the ferrochrome slag powder loses water, which increases the fluidity of the cement mortar [15]. When the substitution amount of ferrochrome slag powder exceeds 15%, due to the large substitution amount of ferrochrome slag powder, the time required for the water absorption–dehydration process of chromite slag powder in cement mortar is increased. The water absorption time of ferrochrome-slag-cement composite mortar increases in the early stage, and the cement mortar maintains high viscosity. After the specimen is formed, the water loss of ferrochrome slag powder increases, which causes segregation and bleeding of the cement mortar. Therefore, adding a certain amount of ferrochrome slag powder can adjust the fluidity of cement mortar. However, when the amount of ferrochrome slag powder is high, cement mortar is prone to segregation and bleeding.

From Figure 8, it can be seen that with an increase in the substitution amount of ferrochrome slag powder, the compressive strength increases and then decreases. When the substitution amount of ferrochrome slag powder is 15%, the compressive strength of the ferrochrome-slag-cement composite rubber material is the highest. In the early stage of hydration, because of the low early hydration of ferrochrome slag powder, the early compressive strength of the cement mortar decreases [16]. In the later stage of hydration, because of the large specific surface area of ferrochrome slag powder, the cement mortar can be filled with cement hydration products. Because of the porous structure of ferrochrome slag powder, the mechanical bite force between the hydration products can be improved. In the later stage of hydration, ferrochrome slag powder can improve the bonding force between hydration products. In addition, due to the hydration reaction of active $SiO_2$ and $Al_2O_3$ in the chromite slag powder with $Ca(OH)_2$ in the later stage of hydration, calcium-aluminum-silicate-hydrate (C-A-S-H) and other hydration products are generated, which increases the compressive strength of the chromite slag powder cement mortar in the later stage [17]. Therefore, according to the test results of fluidity and compressive strength of chromite slag cement composite mortar, a substitution amount of chromite slag powder of 15% leads to the best performance for ferrochrome-slag-cement composite material.

### 3.2. SEM and XRD Analysis

The two groups of cement mortar with substitution amounts of 0% and 15% were analyzed by SEM and XRD to study the effects of the substitution amount of ferrochrome slag powder on the microstructure of cement mortar. The results are presented graphically in Figures 9 and 10.

It can be seen from Figures 9 and 10 that the reference mix hydration products were mainly flocculent C-S-H gel, Aft, and layered $Ca(OH)_2$. Since the hydration products generated after the hydration reaction of cement particles reduce its volume, there are small microcracks and pores inside the cement mortar in the reference mix. When the replacement amount of ferrochrome slag powder is 15%, the partially active $Al_2O_3$ and $SiO_2$ in the ferrochrome slag powder react with $Ca(OH)_2$ to form C-S-H and C-A-S-H gels, and the gel and ettringite crystals intertwine to form a dense network structure. The structure can effectively fill the pores between hydration products. Furthermore, due to the large specific surface area of ferrochrome slag powder, it can fill the gaps between hydration products and between hydration products and cement particles. Moreover, ferrochrome

slag powder has a porous structure, which can improve the mechanical bite force and bonding force between hydration products, to increase the density of the microstructure of ferrochrome-slag-cement composite mortar.

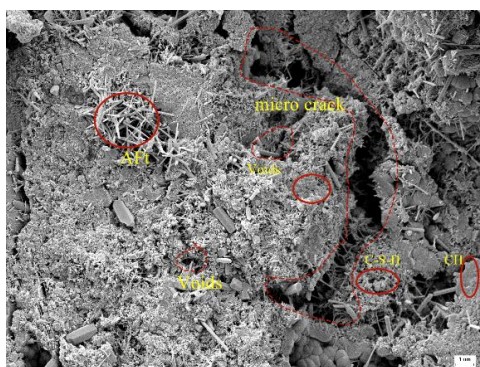

**Figure 9.** SEM photograph of 0% ferrochrome slag powder.

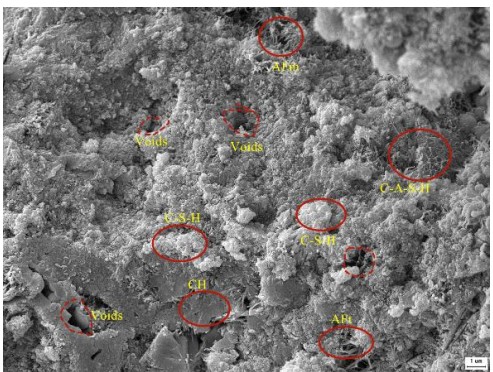

**Figure 10.** SEM photograph of 15% ferrochrome slag powder.

It can be seen from Figure 11 that the main hydration products of cement mortar are C-S-H gel, ettringite (AFt), AFm, Ca(OH)$_2$, and cement particles at 28 d of hydration. When the substitution amount of ferrochrome slag powder is 15%, the obvious diffraction peaks in the hydration products are mainly due to dissolution of the active components in the ferrochrome slag powder, which increases the concentration of AlO$^{2-}$ in the hydration system and promotes the formation of C-A-S-H.

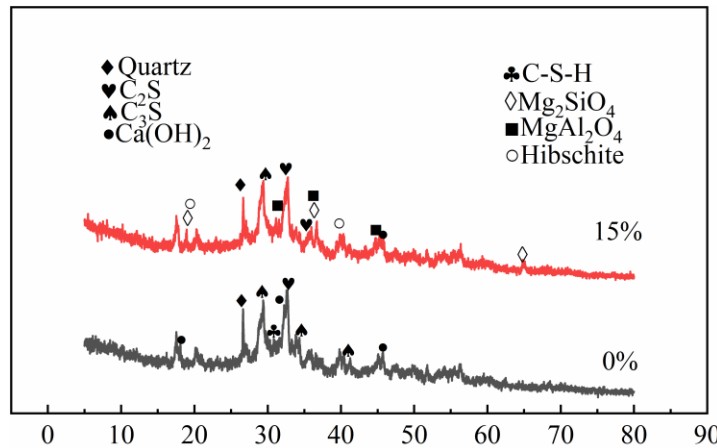

**Figure 11.** XRD of ferrochrome slag cement mortar.

C-A-S-H formed by the hydration reaction reduces the content of AlO$^{2-}$ and Ca$^{2+}$ in the system, which promotes the hydration reaction between the active components in

the ferrochrome slag powder and Ca(OH)$_2$ to generate C-S-H and C-A-S-H gels. At the same time, the decrease in Ca$^{2+}$ concentration in the system also promotes the full reaction of C$_2$S/C$_3$S without hydration to produce more C-S-H gels, filling the gaps between the hydration products, and making the microstructure of cement mortar denser, thereby improving the compressive strength of the cement mortar. When the substitution amount of chromite slag powder is 15%, phase analysis of the ferrochrome-slag-cement composite rubber material reveals the diffraction peaks of forsterite and spinel. This indicated that the Mg element in the chromite slag powder mainly exists in the cement mortar system in a structurally stable forsterite and spinel mineral phase, because the structure of forsterite and spinel is relatively stable in the natural environment.

### 3.3. Study of Basic Properties of Ferrochrome-Slag-Based Low-Carbon Concrete

#### 3.3.1. Slump and Compressive Strength

Ferrochrome slag sand was used to replace 40% of the natural sand, and crude ferrochrome slag completely replaced the natural coarse aggregate. Ferrochrome-slag-based low-carbon concrete with a strength grade of C50 was prepared. CG0 was the control concrete, CG1 was ferrochrome slag aggregate concrete, and CG2 was composite ferrochrome slag aggregate concrete in which ferrochrome slag powder replaced 15% of the cement. The effects of the ferrochrome slag powder and ferrochrome slag aggregate on the compressive strength of concrete were studied.

From Table 5, it can be seen that the slump of ferrochrome slag concrete in CG1 and CG2 is lower than that in CG0, because the ferrochrome slag aggregate has a porous structure and the weight of ferrochrome slag itself is higher than that of natural aggregate. In the process of concrete mixing, the water absorption-loss process of the ferrochrome slag aggregate, the water absorption of ferrochrome slag has an adverse effect on the workability of the concrete mixture, and the porous structure of ferrochrome slag reduces the fluidity of the slurry. Therefore, more slurry is required to wrap aggregate particles to improve the workability of the concrete, so that the density of ferrochrome slag concrete in CG1 and CG2 is higher than that in CG0, because the weight and specific gravity of ferrochrome slag itself is higher than that of natural aggregate. This leads to a higher average density for ferrochrome slag concrete than conventional concrete. It can be seen from Figure 12 that the early strength of CG1 and CG2 were 12.5% and 5.7% higher, respectively, than that of CG0. At 28 d of curing, the compressive strength of CG1 and CG2 were 15.8% and 3.6% higher, respectively, than that of CG0. At 56 d of curing, the compressive strength of the CG0, CG1, and CG2 concretes were 1.5, 2.9, and 3.6 MPa higher, respectively, then those at 28 days. It can be seen that under the same water–binder ratio, the use of ferrochrome slag as concrete aggregate can improve the compressive strength of concrete [18–20].

**Table 5.** Mix proportion of ferrochrome slag concrete.

| Specimen Code | OPC | FA | Fine Aggregate | | CA | Water | PC | Slump | Density |
|---|---|---|---|---|---|---|---|---|---|
| | | | FeCr | NFA | | | | | |
| CG0 | 345 | 60 | | 770 | 1114 | 160 | 8.1 | 205 | 2484 |
| CG1 | 345 | 60 | 308 | 462 | 1114 | 160 | 8.1 | 200 | 2537 |
| CG2 | 345 (15% FeCr) | 60 | 308 | 462 | 1114 | 160 | 8.1 | 190 | 2553 |

The early compressive strength of CG1 and CG2 was higher than that of CG0. The main reason for this was that ferrochrome slag has a porous structure, which can increase the mechanical bite force between the slurry and aggregate, and improve the interface bonding structure between the slurry and aggregate. The hardness of ferrochrome slag is higher than that of traditional natural aggregate [21,22]; therefore, ferrochrome slag as aggregate can improve the early compressive strength of concrete. With an increase in the curing time, the late strength development rate of ferrochrome slag concrete is higher than that of traditional concrete [23], and the strength increase of CG2 was greater than that of

CG1. When ferrochrome slag is used as concrete aggregate, there is a water absorption-loss process, due to its porous structure and an increase in the curing time. The gradual water loss of ferrochrome slag aggregate makes the cement paste around the aggregate fully hydrated, causing more hydration products to fill the internal cracks of the concrete and improve the compressive strength of the concrete in the later stage. In CG2, due to the partial ferrochrome slag powder in the cementitious material, the active components in the ferrochrome slag powder undergo a secondary hydration reaction in the later stage of hydration, and the hydration products fill the microcracks between the slurry and the aggregate, making the concrete denser. This further improves the compressive strength of the concrete [24–26]; therefore, the later compressive strength growth rate of CG2 concrete was higher than that of the other two groups.

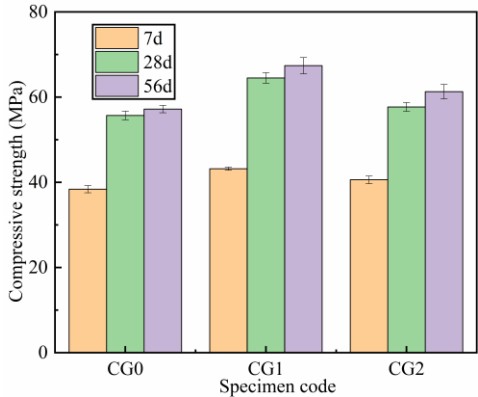

**Figure 12.** Compressive strength of ferrochrome slag concrete.

### 3.3.2. Water Absorption

Water absorption is an important index to measure the durability of hardened concrete, and reducing the water absorption of concrete can significantly improve the long-term performance of concrete. Therefore, to study the effect of ferrochrome slag as aggregate on the long-term durability of concrete, the water absorption changes of the three groups of concrete were explored. The results are presented in Figure 13.

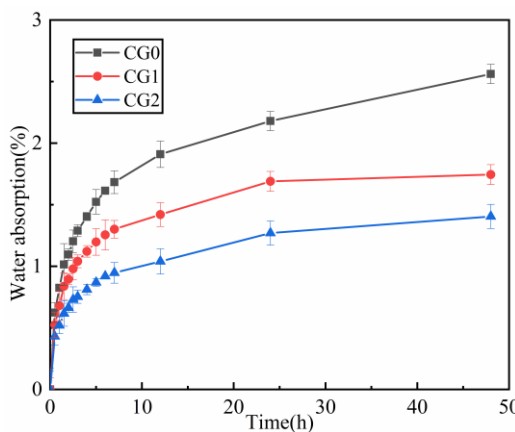

**Figure 13.** Water absorption of ferrochrome slag concrete.

From Figure 13, it can be seen that the water absorption rate of the three groups of concrete increased with time and that the water absorption rate of CG0 was higher than that of CG1 and CG2. That of CG2 was the lowest because of the porous structure of ferrochrome slag as concrete aggregate. The concrete aggregate can improve the bonding structure between aggregate and slurry [27]. In the later stage of curing, the cement slurry can be fully hydrated; therefore, the interior of the concrete becomes denser, thus reducing the water absorption rate of concrete. In CG2, the ferrochrome slag powder has replaced

15% of the cement. The specific surface area of the ferrochrome slag powder is greater than that of the cement; therefore, the gap between cement hydration products and aggregates can be filled, the number of microcracks in the concrete can be reduced, and the water absorption rate of the concrete can be reduced. In the later stage of hydration, some active components in ferrochrome slag powder undergo secondary hydration. The hydration products fill the capillary pores inside the concrete, improve the microscopic pore structure inside the concrete, and reduce the water absorption rate of the concrete. Studies have demonstrated that the greater the water absorption rate of concrete is, the more capillary pores are inside the concrete and the lower the compressive strength is. Therefore, the use of ferrochrome slag as concrete aggregate can reduce the water absorption rate and further improve the compressive strength of concrete.

### 3.3.3. SEM Analysis

To study the interface bonding between ferrochrome slag and cement paste when chromite slag is used as aggregate, and the influence of chromite slag powder on the microstructure of concrete, the microstructure of CG0 and CG2 concretes was analyzed.

As illustrated in Figure 14, the SEM analysis of the concretes at different curing ages indicated that the hydration reaction occurs in the cement paste of CG0 and CG2. The hydration products are wrapped around the aggregate, which increases the bonding force between the concrete paste and the aggregate, thereby improving the compressive strength of the concrete [28]. The interface bonding between the concrete aggregate and the slurry of CG2 with chromite slag as aggregate is denser than that of CG0. The number and width of microcracks between the chromite slag and slurry are smaller than those of traditional concrete, due to the use of chromite slag as aggregate [29]. The porous structure increases the mechanical bite force between the slurry and aggregate. Moreover, ferrochrome slag loses water in the later stage of curing, which further hydrates the cement paste around the aggregate. Hydration products fill the microcracks between the aggregate and slurry, making the aggregate and slurry denser. The compressive failure of concrete occurs at the interface bonding between the aggregate and slurry. The secondary hydration reaction of ferrochrome slag powder and $Ca(OH)_2$ occurs in the later stage of curing, which reduces the content of $Ca(OH)_2$ at the interface bonding between the aggregate and slurry. The hydration products are wrapped around the aggregate, which improves the interface bonding force between the aggregate and the slurry, thereby improving the compressive strength of the concrete.

### 3.3.4. TG-DSC Analysis

To study the change in the number of amorphous structures in the hydration products of ferrochrome-slag-based concrete, two groups of hardened concrete pastes were analyzed by TG-DSC, as illustrated in Figure 15.

From Figure 15, it can be seen that the two groups of samples can be divided into four stages: The first stage is from 30 °C to 200 °C. In this stage, mass loss is mainly caused by the evaporation of pore water and the endothermic decomposition of C-S-H gel, AFt, and AFM. At approximately 120–200 °C, the absorption peaks of AFt, AFM, and C-S-H gel are observed. The mass loss of CG0 is 7.28% higher than that of CG2, and the substitution of cement reduces the content of $C_2S$ and $C_3S$ in the system, resulting in the reduction of hydration products. The corresponding temperatures of the AFt and AFm absorption peaks in CG2 are higher than those in CG0. C-A-S-H gel is produced by the reaction of $AlO^{2-}$ with the hydration products of cement in ferrochrome slag powder. $AlO^{2-}$ is wrapped between AFt and AFm to increase the decomposition temperature. The second stage is 200 °C–410 °C. This stage involves mass loss of type II C-S-H and C-A-S-H gels caused by thermal dehydration. Here, the mass loss of CG2 gel is greater than that of CG1 gel, and the strength of the DSC absorption peak is weaker, due to substitution of a small amount of ferrochrome slag powder in which there is partial active $AlO^{2-}$ dissolution, where one part promotes AFt-to-AFm transformation while the other part reacts with cement hydration

products to produce C-A-S-H gel. The third stage is from approximately 420 °C to 600 °C. This stage mainly involves mass loss caused by the decomposition of CH. At the same time, some carbonate and C-S-H are decomposed in the later stage. The mass loss of CG2 is 0.81% lower than that of CG0. According to the DSC curve, the intensity of the diffraction peak produced by thermal decomposition of CH in CG2 is weaker than that of CG0, and the results show that the content of CH in CG2 is lower than that in CG0. The fourth stage is from approximately 610 °C to 800 °C. According to the DSC curve, the absorption peak of CG2 is stronger in this stage, which is mainly the recrystallization temperature of the new phase produced by the decomposition and hydration of carbonate. The results indicate that the hydration gel of CG2 dehydrates and transforms into mineral phases such as β-wollastonite [30,31], and the hydration products absorb $CO_2$ from the environment to produce calcite and carbonate phases, so there is a strong absorption peak.

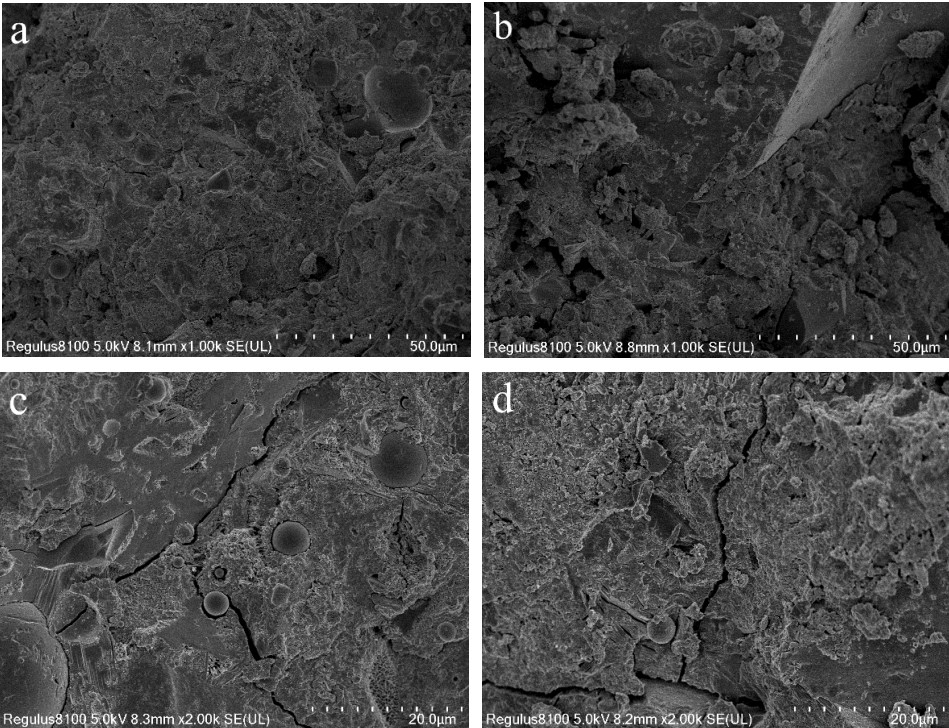

**Figure 14.** SEM of ferrochrome slag concrete ((**a**). CG0 curing 7 d; (**b**). CG2 curing 7 d; (**c**). CG0 curing 28 d; (**d**). CG2 curing 28 d).

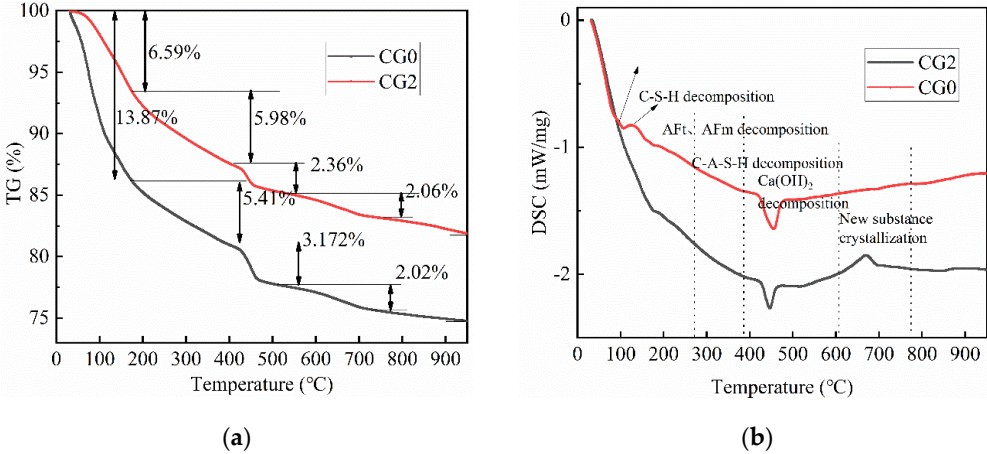

**Figure 15.** TG-DSC curves for ferrochrome slag concrete: (**a**) TG curve; (**b**) DSC curve.

According to the analysis of the microstructure of ferrochrome-slag-based concrete, when ferrochrome slag is used as concrete aggregate, because of its porous structure, the interfacial bond between aggregate and paste can be increased. Moreover, ferrochrome slag powder can fill microcracks and rehydrate the concrete with the hydration products of cement, which can reduce internal cracks and improve the compressive strength of concrete.

## 4. Conclusions

(1) The effect of ferrochrome slag powder on the strength and fluidity of cement mortar was examined in this study. It was determined that there is a process of water absorption and loss in cement mortar. Due to the porous structure and water-reducing effect of ferrochrome slag powder, the fluidity of cement mortar can be adjusted and the compressive strength can be improved by adding 15% ferrochrome slag powder.

(2) The microstructure analysis indicated that the dissolution of active components in ferrochrome slag powder promotes the transformation from AFt to AFm. Active $Al_2O_3$ and $SiO_2$ react with $Ca(OH)_2$ to form C-S-H and C-A-S-H gels. The ferrochrome slag powder can effectively fill the pores between the hydration products to improve the compressive strength of mortar. The Mg element in ferrochrome slag powder mainly exists in the cement mortar system as magnesium olivine and spinel mineral phases with a stable structure, which does not cause problems for system stability.

(3) The workability, compressive strength, and water absorption of ferrochrome-slag-based concrete were studied. It was found that ferrochrome slag used as aggregate can reduce the fluidity of concrete under the same mix proportions; however, the compressive strength of ferrochrome-slag-based concrete is higher than that of conventional concrete. Ferrochrome slag can reduce the water absorption of concrete and improve the compressive strength of concrete. This study provides a reference for the application of water-cooled ferrochrome slag as fine aggregate and air-cooled ferrochrome slag as coarse aggregate in practical engineering.

(4) TG-DSC and SEM analyses revealed that when ferrochrome slag is used as concrete aggregate, the interface bonding properties between aggregate and paste can be improved because of the porous structure of ferrochrome slag.

(5) This study found that water-cooled high-carbon ferrochrome slag powder can be used as an auxiliary cementitious material. When the substitution amount of chromite slag powder is 15%, the compressive strength of the ferrochrome-slag-cement composite rubber material is the highest. This study provides a reference for the use of water-cooled high-carbon ferrochrome slag as a supplementary cementitious material in practical engineering.

**Author Contributions:** Conceptualization, M.H. and J.W.; methodology, M.H. and J.W.; software, J.W.; validation, M.H., J.W., X.Z. and M.S.; formal analysis, J.W.; investigation, M.H., J.W., X.Z. and M.S.; resources, M.H.; data curation, J.W.; writing—original draft preparation, J.W.; writing—review and editing, M.H.; visualization, M.H. and J.W.; supervision, M.H.; project administration, M.H.; funding acquisition, M.H. All authors have read and agreed to the published version of the manuscript.

**Funding:** This research was funded by [Natural Science Foundation of China] grant number [No. 52168033] And The APC was funded by [Meiyan Hang].

**Institutional Review Board Statement:** This study did not require ethical issues.

**Informed Consent Statement:** This research did not involve humans.

**Data Availability Statement:** Data is contained within the article, The data presented in this study are available in [Design and study of physical and mechanical properties of concrete based on ferrochrome slag and its mechanism analysis].and further inquiries can be directed to the corresponding authors.

**Conflicts of Interest:** The authors declare that they have no known competing financial interest or personal relationship that could have appeared to influence the work reported in this paper.

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
