# Peer review of "Design and Study of Physical and Mechanical Properties of Concrete Based on Ferrochrome Slag and Its Mechanism Analysis"

_buildings, doi:10.3390/buildings13010054_

Round 1

Reviewer 1 Report

The article provides the necessary amount of scientific research. The consistency and logic of the presentation of the article are at a high level. There are also some comments on the article: 1. Since the authors do not investigate CO2 emissions from concrete during the partial replacement of cement with ferrochrome slag, the use of the term "low carbon concrete" in the title of the article is unreasonable. Suggested title: "Design and study of physical and mechanical properties of concrete based on ferrochrome slag and its mechanism analysis" 2. In the Abstract section, it is necessary to describe the relevance of the research with an emphasis on construction and building materials. It is also recommended to add the numerical values ​​of the obtained experimental results: how much the strength increased and the photoabsorption decreased compared to the reference samples. 3. Since the article is submitted to the Buildings journal, it is recommended to start the Introduction section with problems in construction, and not with environmental aspects. Some information (L27-32) is recommended to be removed. Based on the purpose of the studies, it is not clear what the authors have done that is new vs. previous studies. In general, the Introduction section and the purpose of the research are written from the position of solving environmental problems, therefore, in this form, the manuscript is more suitable for the Environments journal. Therefore, it is recommended that the Introduction section be reworked with more emphasis on concrete, construction, and building materials. 4. It is necessary to decipher the abbreviations FSA (L78) OPC (L81) 5. Table 1. Please explain why the sum of OPC oxides is 94.69% instead of 100%. 6. Table 2. Replace "Mpa" with "MPa" 7. Figures 3, and 4 are recommended to be placed immediately after their mention in the text. 8. Text within L177-186 is recommended to be removed, as the water absorption test is standard and does not require a detailed description (at the discretion of the author). 9. Thermogravimetric and differential scanning calorimetric analyses (section 2.2.3) is a standard method that does not require a description of the essence. It is enough to indicate the device only with which the tests were carried out. Text within L 195-199 is recommended to be removed. 10.L262. The phrase "...the replacement amount of ferrochrome slag powder is 0%" is recommended to be replaced by "reference Mix" 11. In Figures 8, 12, and 13 it is necessary to indicate the errors in the form of bars. 12. Figure 11. The imposition of symbols makes it difficult to understand the information. It is recommended to enlarge the figure and change the notation format. 13. Figure 14 is recommended to be deleted, as it does not carry semantic meaning. 14. Studies need to be supplemented with data on average density

Reviewer 2 Report

In this study, the effect of substituting different amounts of cement with ferrochrome slag micropowder on the strength and fluidity of cement mortar was investigated. The paper is well-organized and includes new contributions. However, a number of issues in the manuscript are expected to be solved. Besides, there are some suggestions and questions that should be addressed.

- It will be more appropriate if you add a paragraph at the end of the introduction section illustrating the layout of the paper.

- Please declare the limitations of the application of ferrochrome slag micropowder.

-The authors should check the cited papers, line 78 “Acharya [10]” is not cited correctly.

-In lines 78 and 81, please introduce “FCA” and “OPC”

-Please introduce Fig. 14 in more detail and give more explanation for the elements which was used in the water absorption test.

-In the conclusion please clearly mention the practical outcome of the study. Please clarify what the employed technique offers to the engineering field compared to methods available in the literature.

Reviewer 3 Report

In this paper, high-carbon ferrochrome slag powder produced by grinding was used to replace different proportions of cement, and the effect of the amount of ferrochrome slag powder on the physical and mechanical properties of ferrochrome-slag-cement composites was analyzed. Overall speaking, this is a very interesting and well-organized paper. The experiments were well designed and the conclusions are well supported by the experiments. Below are my comments.

1. Line 83: it is better to use the well defined terminology "supplementary cementitious material (SCM)" in cement and concrete industry.

2. The knowledge gap has not been clearly explained. How will this study contribute to the current limit understanding of the problem? The objective and innovation of this paper should be emphasized in the last paragraph of introduction.

3. The literature review is insufficient. The SiO2 bearing industrial byproducts can be used as supplementary cementitious materials and also as the raw materials for alkali-activated cements. It is recommended that the authors should add the information about alkali-activated materials (AAMs). The recently published papers should be reviewed. (e.g., "Analytical investigation of phase assemblages of alkali-activated materials in CaO-SiO2-Al2O3 systems: The management of reaction products and designing of precursors. Materials & Design194, p.108975.")

4. The authors should describe the broad hump in the XRD pattern of the slag which shows the amorphous phase and chemical reactivity of the material.

5. The loading rate for the mechanical property tests should be specified.

6. More discussions should be made on the relationship between mechanical properties of the hardened concrete and the microstructure.

7. The conclusions should be rearranged. The authors may use the bullet points to show every conclusion. Also, the conclusion part is a little lengthy.

Round 2

Reviewer 1 Report

The authors have made all the adjustments according to the comments. The manuscript can be accepted in present form.

Reviewer 3 Report

This paper has been revised based on the comments